# Roles for α-Synuclein in Gene Expression

**DOI:** 10.3390/genes12081166

**Published:** 2021-07-29

**Authors:** Mahalakshmi Somayaji, Zina Lanseur, Se Joon Choi, David Sulzer, Eugene V. Mosharov

**Affiliations:** 1Departments of Psychiatry and Neurology, Columbia University Medical Center, New York, NY 10032, USA; ms5335@cumc.columbia.edu (M.S.); zina.lnsr@gmail.com (Z.L.); sjc2167@cumc.columbia.edu (S.J.C.); ds43@cumc.columbia.edu (D.S.); 2Division of Molecular Therapeutics, New York State Psychiatric Institute, Research Foundation for Mental Hygiene, New York, NY 10032, USA

**Keywords:** Parkinson’s disease, α-Synuclein, gene expression, signal transduction, epigenetics, silencing therapeutics, nuclear receptors, calcium channels

## Abstract

α-Synuclein (α-Syn) is a small cytosolic protein associated with a range of cellular compartments, including synaptic vesicles, the nucleus, mitochondria, endoplasmic reticulum, Golgi apparatus, and lysosomes. In addition to its physiological role in regulating presynaptic function, the protein plays a central role in both sporadic and familial Parkinson’s disease (PD) via a gain-of-function mechanism. Because of this, several recent strategies propose to decrease α-Syn levels in PD patients. While these therapies may offer breakthroughs in PD management, the normal functions of α-Syn and potential side effects of its depletion require careful evaluation. Here, we review recent evidence on physiological and pathological roles of α-Syn in regulating activity-dependent signal transduction and gene expression pathways that play fundamental role in synaptic plasticity.

## 1. Molecular Organization of α-Syn and Its Role in Pathology

α-Syn is a 140 amino acid protein (~15 kDa) encoded by the *SNCA* gene on chromosome 4 (Chr4q22.1). In its monomeric state, the protein is soluble and intrinsically unfolded, but has also been identified as a helically folded tetramer under physiological conditions [1]. The protein can furthermore form multimeric structures, including oligomers, fibrils and more complex aggregates under pathological conditions.

The sequence of α-Syn is typically divided into three main regions: N-terminus, C-terminus, and the central non-amyloid-β component (NAC) region. The lysine-rich N-terminus can acquire an amphipathic α-helical structure formed by four repeats of the KTKEGV consensus sequence [2] that are also required for the tetramerization of α-Syn [3]. The α-helices provide for reversible binding of α-Syn to phospholipid membranes, particularly those with high curvature [4], which play critical roles in the control of presynaptic fusion machinery [5,6,7]. Pathological nucleation and aggregation of α-Syn depends on both the N-terminus and the hydrophobic NAC region that can form β sheets [8]. All of the known *SNCA* familial PD mutations to date—A30P, E46K, H50Q, G51D, A53E and A53T—are found in the N-terminus domain [9,10].

The C-terminus is comprised of an EF-hand-like sequence that is negatively charged and is typically unstructured [11]. It is capable of binding Ca^2+^ and oxidized dopamine (DA) that affect both normal and pathological properties of the protein. The binding of oxidized DA to α-Syn is non-covalent, reversible, and occurs at the Y125EMPS129 pentapeptide of the C-terminal region with an additional long-range electrostatic interaction with E83 in the NAC region [12,13]. DA-modified α-Syn is less likely to fibrilize and instead forms soluble oligomers that are cytotoxic [14,15]. Ca^2+^ binding increases the ability of α-Syn to bind synaptic vesicle membranes [16], at the same time affecting protein ligand binding and oligomerization [17]. Interestingly, overexpression of truncated α-Syn that lacks the C-terminus is sufficient to elicit a PD-like phenotype in mice [18], suggesting that it modulates rather than drives α-Syn-mediated toxicity. Finally, C-terminus harbors a phosphorylation site at serine 129 (pS129). While the physiological role of this post-translational modification is currently unknown, its accumulation under pathological conditions suggests that it may play a role in α-Syn oligomerization or toxicity [19]. 

### 1.1. Synucleinopathies

All “synucleinopathies” are characterized by the accumulation of insoluble aggregates mainly comprised of amyloid fibrils of phosphorylated α-Syn. In Parkinson’s disease (PD), PD with dementia, dementia with Lewy bodies, and pure autonomic failure, the inclusions (termed Lewy bodies and Lewy neurites) are found in peripheral and central neurons [20,21], whereas in multiple system atrophy, the argyrophilic glial cytoplasmic inclusions (also known as Papp-Lantos bodies) are detected inside oligodendrocytes [22].

PD is the most prevalent synucleinopathy, affecting ~1% of the population over the age of 60 [23]. This progressive neurodegenerative disorder is characterized by four cardinal symptoms—tremor, rigidity, bradykinesia, and postural instability—and a plethora of secondary symptoms that affect both central and peripheral systems. Neuropathologically, the feature that correlates with the cardinal PD symptoms is neuronal loss of pigmented DA neurons of the substantia nigra pars compacta with concomitant dopaminergic denervation of the striatum [24]. 

The α-Syn protein plays critical roles in both sporadic and familial PD as (1) mutations or multiplications of the *SNCA* gene cause autosomal-dominant PD [25], (2) genome-wide association studies show a correlation between single-nucleotide polymorphisms in the *SNCA* locus and the risk of developing sporadic PD [26,27], (3) levels of phosphorylated α-Syn are increased in post-mortem brains of PD patients and patient-derived dopaminergic neurons [28], (4) DA neurons that lack α-Syn are protected in neurotoxin and genetic models of PD [29,30,31]. Additionally, α-Syn has been shown to interact with several proteins implicated in PD and other neurodegenerative disorders, including tau [32,33], LRKK2 [34], Fyn [35], parkin [36], and DJ-1 [37].

α-Syn pathology, however, occurs throughout the nervous system in PD and Lewy bodies disease patients, and in some cases does not correlate with cell death [38], suggesting that the protein is necessary but not sufficient for neurodegeneration. Additionally, some PD cases, in particular those associated with LRRK2 and parkin genes mutations, do not exhibit Lewy body pathology [39,40].

### 1.2. Therapeutic Silencing of α-Syn 

Due to the significance of α-Syn in PD pathology, recent advances in therapeutic approaches intended to reduce patient α-Syn levels have gained momentum, including the use of antibodies, vaccines, antisense-oligonucleotides, and small molecules [41,42]. Preclinical studies in cell cultures and animal models have demonstrated successful *SNCA* gene silencing using antisense oligonucleotides (ASOs), ribozymes, siRNAs, or miRNA without neurotoxic effects [43,44,45,46,47,48,49,50]. Furthermore, several small molecules aimed at preventing α-Syn aggregation (developed by UCB, Proclara and Sun pharmaceutical companies) as well as passive and active immunotherapies that target monomeric or aggregated α-Syn (Prothena, Biogen, AstraZeneca, Takeda, Affiris, Endobody) are currently in clinical trials (clinicaltrials.gov). 

With the possible exception of immunotherapies that target pathological oligomeric and misfolded forms of α-Syn, such as cinpanemab [51], prasinezumab [52], and ABBV-0805 (currently in phase I clinical trial), these approaches are expected to deplete monomeric α-Syn. As this protein is abundantly expressed in the brain and the periphery, changes in normal function and potential side effects of decreased α-Syn levels need careful evaluation. For example, multiple studies demonstrate roles for α-Syn in regulating dopamine neurotransmission both ex vivo (reviewed in [6]) and in anesthetized animals in vivo [53,54], indicating that α-Syn deficiency might lead to synaptic dysfunction. Although α-Syn knockout mice (α-Syn^−/−^) are viable and fertile, these animals exhibit decreased behavioral responses to amphetamine [55] and reduced learning ability in tests requiring both working and spatial memory [56]. Moreover, despite the evidence for neuroprotection in α-Syn^−/−^ mice, acute α-Syn knock-down in adult rats unexpectedly resulted in neuroinflammation and degeneration of nigral dopaminergic neurons [57,58,59]. It may be that the developmental stage in which α-Syn depletion occurs is important, or additional mechanisms govern protective or deleterious consequences of protein expression. In either case, these contrasting reports highlight the importance of understanding the physiological role of the protein and the consequences of its decreased expression, particularly in mature animals.

## 2. Role of α-Syn in Regulating Gene Expression

α-Syn is predominantly localized at presynaptic terminals where it is involved in regulating synaptic vesicle exocytosis [54,60,61,62,63,64]. Additionally, α-Syn can be found in other cellular compartments, including the nucleus [65,66,67], mitochondria [68,69], endoplasmic reticulum [70,71,72], and Golgi apparatus/lysosomes [73,74,75]. Besides lipid membranes and PD-related proteins mentioned above, molecular partners of α-Syn include 14-3-3 chaperon protein [76], Hsp70-interacting protein (CHIP) [77], synphilin-1 [78], tubulin [79], calmodulin [80,81], and others [82,83]. 

Such promiscuity of α-Syn localization and interactions makes it difficult to isolate a single mechanism of action. While cellular organelles and pathways involved in α-Syn physiological function are highlighted in several recent reviews [6,83], here we focus on an aspect of α-Syn normal function that has been relatively ignored—the role of the protein in transcriptional control and immediate early genes induction in neurons. Several lines of indirect evidence support the pathophysiological relevance of α-Syn in transcription. Microarray analysis of gene expression in neuroblastoma cell lines transfected with WT α-Syn revealed 15 genes that were upregulated and 30 that were downregulated [84]. Similarly, RNA-seq analysis of induced pluripotent stem cells (iPSC)- derived dopaminergic neurons from PD patients carrying a triplication of the *SNCA* locus identified 131 upregulated and 115 downregulated genes compared to neurons from healthy control [85]. While, unsurprisingly, some of these changes can be attributed to stress response and other processes downstream of α-Syn-induced aggregation and toxicity, there is evidence supporting a role for the protein in directly affecting gene expression as discussed below. 

### 2.1. Interaction of α-Syn with DNA and Histones 

α-Syn has been reported to directly interact with DNA, RNA-interacting proteins, and histones. Since this topic has been recently thoroughly reviewed [86,87], we only provide a brief overview updated with recent findings. 

#### 2.1.1. Direct DNA Binding

Unmodified [88] and glycated [89] α-Syn can bind to supercoiled DNA, leading to changes in DNA conformation and stability that in turn can modulate α-Syn conformation and function [90]. Binding of α-Syn to DNA with concomitant changes in gene expression was recently confirmed with chromatin immunoprecipitation followed by next-generation sequencing (ChIP-seq) technique in Lund human mesencephalic (LUHMES) cells and transgenic mice expressing α-Syn [91]. Nuclear localization of α-Syn in these models depended on its phosphorylation at serine 129 and provided neuroprotection compared to the cytosolic protein, providing a potential link between the subcellular localization, post-translational modification (PTM), and toxicity of α-Syn. 

As an alternative, α-Syn may prevent DNA damage by colocalizing with DNA damage response components [92]. α-Syn knock-out mice showed increased neuronal double strand breaks that were rescued by re-expression of human α-Syn. The authors propose that cytoplasmic aggregation of α-Syn decreases its nuclear translocation, in turn increasing DNA damage, and this nuclear loss-of-function might trigger pro-apoptotic signaling. Interestingly, α-Syn has been also predicted to bind its own mRNA in an autogenous way, which might be instrumental for the regulation of protein translation through a negative feedback mechanism [93]. 

In contrast to direct binding to polynucleotides, α-Syn may affect gene expression by altering DNA methylation. Post-translational modification of DNA and histones are key mechanisms of epigenetic control of gene expression that in addition to playing a role in synaptic plasticity and development are involved in multiple neurodegenerative disorders [94,95]. Typically, methylation of CpG-rich promoter regions by DNA methyl transferases (DNMT) represses gene transcription, which is countered by passive DNA demethylation that restores promoter activity [96]. Association of α-Syn with DNMT1, an enzyme that is normally located in the nuclei, causes its extrusion to the cytosol, resulting in global DNA hypomethylation (Figure 1). In human and mouse brain, this was shown to affect the expression of multiple genes including *SNCA*, *SEPW1*, and *PRKAR2A* [97]. 

#### 2.1.2. Interaction with Histones

It has been reported that α-Syn binds to histones under physiological conditions and this interaction increases in neurons exposed to toxic insults and promotes α-Syn aggregation [98,99]. Although early reports suggested that nuclear localization and interaction with histones could play a role in α-Syn physiology and toxicity, the molecular changes that result from this interaction have been examined only recently.

The two most common histone PTMs—acetylation and methylation—play key roles in many fundamental cellular processes including transcriptional regulation and chromatin remodeling. Acetylation of histones results in a loose chromatin state that allows gene transcription, whereas histone deacetylation results in a tight chromatin structure that suppresses transcriptional activity. These reactions are catalyzed by families of enzymes that possess histone acetyltransferase (HAT) or histone deacetylase (HDAC) activity. The balance between the activities of HAT and HDAC is tightly controlled in healthy cells but can be disrupted in diseases including PD [100,101,102,103].

Several studies report that α-Syn can reduce histone acetylation, thus inhibiting gene expression (Figure 1). Using α-Syn constructs targeted to specific cellular compartments, it was demonstrated that protein with the nuclear localization sequence causes increased neurotoxicity in SH-SY5Y cells and drosophila brain, whereas targeting α-Syn to the cytosol provides neuroprotection. Nuclear α-Syn was found to directly interact with histone 3 (H3), decreasing its acetylation, whereas HDAC inhibitors protected against α-Syn-mediated neurotoxicity [104]. Protection against α-Syn-mediated toxicity by HDAC inhibition was also shown in human neuroglioma cells and in a drosophila model of Parkinson’s disease where genetic or pharmacological blockade of sirtuin 2, a member of Class-III HDAC family, showed dose-dependent rescue from α-Syn-induced toxicity, supporting a role of α-Syn in histone epigenetic control of gene expression [105]. Similarly, decreased H3 acetylation and altered RNAseq gene expression profiles were found in LUHMES dopaminergic cells overexpressing α-Syn; inhibition of HDAC with sodium butyrate prevented α-Syn-induced DNA damage, apparently via upregulation of genes involved in DNA repair [106]. In a recent study, Mazzocchi and collegues used gene co-expression analysis to demonstrate that HDAC5 and HDAC9 are expressed in dopaminergic neurons in the human and mouse substantia nigra where they act as negative regulators of the BMP-Smad pathway, thus limiting the extent of neurite growth [107]. Genetic or pharmacological targeting of these class-IIa HDACs promoted neurite growth in cells overexpressing α-Syn and protected cultured rat DA neurons against a parkinsonian neurotoxin, MPP+. In contrast to these studies however, selective inhibition of class III HDACs with nicotinamide dose-dependently exacerbated degeneration of DA neurons in lactacystin-lesioned rats [108], highlighting the importance of targeting specific HDACs in possible therapeutic applications. Although pharmacological inhibition of HDAC activity is currently under investigation as disease modifying PD therapy [100,109], some of their effects may be mediated by mechanisms unrelated to histone acetylation, such as microtubule stabilization [110,111]. 

HAT activity may be directly affected by α-Syn. Cytosolic α-Syn decreases the levels of p300, a neuronally expressed HAT both in vitro and in vivo [112]. Besides reduced histone acetylation, diminished activity of p300 also leads to decreased acetylation of p65 thereby blocking the nuclear factor-κB (NF-κB) binding to the protein kinase C delta (PKCδ) promoter region. Thus, in this model, the presence of α-Syn confers protection due to transcriptional suppression of PKCδ, inhibiting its proteolytic activation that would otherwise activate caspase 3-mediated proapoptotic signaling in DA neurons. 

Whereas histone acetylation is thought to always promote gene expression, histone methylation regulates transcription depending on the context. For example, H3 methylation of lysine at position 4 (H3K4 modification) activates the nearby DNA loci, whereas modifications of the same histone at positions 9 or 27 (H3K9 and H3K27) inactivate gene expression. 

A significant decrease in H3K36 double-methylation was reported in an α-Syn yeast proteinopathy model, together with milder changes in other histone PTM [113]. In contrast, overexpression of α-Syn in transgenic drosophila and SH-SY5Y neuroblastoma cells upregulated euchromatic histone lysine N-methyltransferase 2 (EHMT2) activity by a retinoic acid (RA)-dependent mechanism [114]. Interaction of EHMT2 with the repressor element-1 (RE1)-silencing transcription factor (REST) led to transcription inactivation via EHMT2-mediated H3K9 mono- and di-methylation, silencing the expression of *SNAP25* and *L1CAM* genes in cells overexpressing α-Syn (Figure 1).

### 2.2. α-Syn-Dependent Regulation of Nuclear Receptors

The capacity of higher organisms to learn from previous experience in order to adapt their behavior depends on the ability of neurons to convert transient stimuli into long-lasting alterations in structure and function. One such synaptic plasticity mechanism involves activity-dependent upregulation of gene expression via nuclear translocation of nuclear receptors and transcription factors. These proteins typically reside in the cytoplasm where they undergo phosphorylation or dephosphorylation following G protein-coupled receptors activation or activity-dependent opening of voltage-gated Ca^2+^ channels. Once activated, they are translocated into the nucleus where they bind to the promoter regions and affect the transcription of a variety of genes responsible for proliferation, differentiation, survival, long-term synaptic potentiation, neurogenesis, and neuronal plasticity. 

Retinoic-acid receptor (RAR) and peroxisome proliferator-activated receptor-γ (PPARγ) are members of the nuclear-receptor superfamily that bind to DNA as heterodimers with retinoid-X receptors (RXRs). Using SH-SY5Y cells and primary neuronal cultures, a recent study showed that α-Syn binds to retinoic acid, followed by nuclear translocation of the complex via a calcium- and calreticulin-dependent process [115]. Once in the nucleus, α-Syn/RA activates RAR/RXR- and PPARγ/RXR- dependent gene transcription via binding to the corresponding response elements on the DNA (Figure 2). Nuclear α-Syn translocation was shown to occur at physiological levels but also increased if the protein was overexpressed or mutated, providing a possible link between a normal role of α-Syn in the regulation of RA-mediated gene transcription and its toxicity in synucleinopathies [115,116]. Interestingly, another study from the same group reported that α-Syn expression downregulates PPARγ levels and activity in a transgenic mouse model of PD and a cell culture system. This in turn decreases catalase expression and compromises cell survival: an effect rescued by pharmacological activation of PPARγ, PPARα, or RXR [117]. On the other hand, activation of these nuclear receptors enhanced the accumulation of soluble α-Syn oligomers, pointing to a possible balance between the physiological and pathological roles of α-Syn. 

Analysis of genes regulated by nucleus-targeted α-Syn using RNAseq in primary dopamine neuronal cultures revealed that Nurr1—another nuclear receptor that forms heterodimers with RXR—is significantly downregulated [118]. Forced Nurr1 expression or activation of Nurr1/RXR heterodimer with an RXR ligand in cells subjected to α-Syn toxicity efficiently rescued the expression of hundreds of dysregulated genes, suggesting that a substantial portion of α-Syn-mediated toxicity may be due to downregulation of Nurr1 (Figure 2). A decrease in Nurr1 expression was also observed in neuroblastoma cell lines transfected with wild-type α-Syn [84] and in human PD [119], implicating this protein in the development and progression of synucleinopathies and prompting a search for clinically applicable agonists of Nurr1/RXR heterodimer as a PD therapy [120,121].

A functional partner of Nurr1 in the adaptive response to stress is peroxisome proliferator receptor γ coactivator-1 (PGC-1α) which, in addition to being a key transcriptional regulator of energy metabolism, functions as a suppressor of reactive oxygen species in neurons. Oxidative stress causes nuclear translocation of α-Syn where the protein binds to the promoter element of PGC-1α (Figure 2), resulting in decreased expression of this transcription factor and downstream mitochondrial genes [122]. PGC-1α levels are also downregulated in post-mortem PD substantia nigra neurons [122,123], whereas exogenous PGC-1α expression protects against α-Syn-mediated toxicity in cellular PD models [124]. We note however that both Nurr1 and PGC-1α are downstream of cyclic-AMP-response-element-binding protein (CREB) and extracellular signal-regulated kinase (ERK) signaling (see below) in dopaminergic neurons [125], and so the mechanism of their downregulation may be more complex. For example, a recent study using primary cell cultures, rodent models, and human post-mortem brains from LBD patients identified a potential mechanism of α-Syn toxicity related to the presence of α-Syn oligomers in the mitochondria [126]. The authors found that decreased expression and activity of SIRT3, a mitochondrial sirtuin that maintains mitochondrial function and prevents oxidative stress [127], lead to decreased CREB phosphorylation. As pCREB controls PGC-1α expression [128], thus affecting mitochondrial dynamics and function [126], and the fact that PGC-1α may also control SIRT3 expression [129], demonstrate the intricacy of cellular RedOx homeostasis, where α-Syn might play both physiological and pathological functions.

### 2.3. α-Syn-Dependent Regulation of Immediate Early Genes

Multiple studies have demonstrated that α-Syn can regulate the expression of immediate early genes, including CREB, nuclear factor of activated T cells (NFAT), and ETS Like-1 protein (Elk1), although the precise mechanisms remain under investigation. Specifically, α-Syn might be involved in at least two signal transduction cascades: MAPK/ERK and Ca^2+^/CAMKII- mediated transcriptional control.

#### 2.3.1. Regulation of Transcription via MAPK/ERK Pathway 

Mitogen-activated protein kinase (MAPK) signaling cascade includes a group of kinases and phosphatases that mediate the nuclear translocation of several transcription factors, including CREB and Elk1. Elk1 is a member of the ternary complex factor (TCF) of transcription factors that includes nuclear phosphoproteins involved in regulating immediate early gene expression. Elk1 in its inactive form is colocalized in the cytoplasm with mitochondrial proteins or microtubules. Elk1 phosphorylation in response to kinases such as extracellular signal-regulated kinase 2 (ERK2) results in its translocation to the nucleus, where pElk1 is implicated in regulating chromatin remodeling and neuronal differentiation. Both ERK2 and Elk1 have been shown to form complexes with WT [76,130,131] and A53T mutant α-Syn [130,132]. This binding reduces the amount of available active ERK2 and Elk1, resulting in altered gene expression and accelerated cell death (Figure 3). However, other reports either did not find a change in pERK levels in cultured rat primary neurons that overexpress α-Syn [133,134], or found an increase in pERK in stable neuronal cells treated with non-toxic levels of extracellular α-Syn [135]. 

#### 2.3.2. Regulation of Transcription via LTCC/CaMK Pathway

Activity-dependent opening of voltage-gated calcium channels provides another mechanism that allows to encode acute information flow into changes in gene expression that form the basis of neural plasticity. In particular, L-type calcium channels (LTCC) are uniquely positioned to couple neuronal activity and immediate early genes expression (Figure 3) [136,137]. 

LTCC are comprised of four subunits that form a hetero-oligomeric complex [138]. LTCC expressed in the brain are represented by Ca_v_1.2 and Ca_v_1.3 types that express α1C and α1D pore-forming subunits, correspondingly. LTCC gating properties are intricately regulated via structural changes of the α1 subunit, whereas auxiliary α2, β, and δ subunits play structural and modulatory roles. Binding of various proteins as well as Ca^2+^-induced phosphorylation/dephosphorylation events on the α1 subunit C-terminus modulate LTCC conductance by inducing Ca^2+^-dependent inhibition and facilitation (CDI and CDF), thus changing the kinetics of channel activation, inactivation, and reactivation [139].

Voltage-dependent opening of the LTCC initiates a signaling cascade when Ca^2+^ binds to calmodulin (CaM), which is constitutively attached to the channel’s α1 subunit [140,141]. Ca^2+^/CaM plays a principal role in this pathway due to its kinase activity as well as its formation of a complex that activates calcineurin (CaN a.k.a. PP2B) phosphatase. Sequential activation of Ca^2+^/CaM-dependent protein kinases (CaMK), including CaMKIIα/β/γ, CaMKIV, and CaMKK [142] ultimately phosphorylates and activates nuclear CREB [143], which binds to a highly conserved cAMP response elements (CRE) on DNA, thereby modulating gene transcription. Similarly, CaN-dependent dephosphorylation of cytosolic NFAT (NFATc) allows its nuclear translocation, where together with nuclear partner proteins (NFATn) that are regulated by other pathways including the MAPKs, it binds to corresponding NFAT response elements [144,145,146,147].

Multiple studies demonstrate an interaction between α-Syn and the members of the LTCC/CaMKII pathway (Figure 3). α-Syn-binding partners were recently studied in primary rat cortical neurons by an ascorbate peroxidase (APEX) assay [148], which is based on the short-lived radicals that label amino acids in their immediate proximity (<10 nm). In addition to α-Syn ligation with members of endosomal pathways, pre-synaptic machinery and mRNA-binding proteins, the assay identified its proximity to CaN catalytic subunits (PPP3CA and PPP3CB), LTCC β subunits (CACNB1 and CACNB3), and serine/threonine-protein phosphatase PP1 catalytic subunits (PPP1CA, PPP1CB, and PPP1CC). 

More directly, it was shown that α-Syn activates CaN and induces NFATc3 nuclear translocation in vitro, in HEK293 cell lines and in primary cultures of DA neurons, [149]. In a follow-up study, the same group investigated the interaction of α-Syn and CaN in vitro using a series of biochemical techniques, including microscale thermophoresis, GST pulldown, and co-immunoprecipitation [150] and reported that α-Syn binds the catalytic subunit of CaN with a Kd of 6.9 ± 0.4 µM, well within the estimated range of cytosolic α-Syn levels [151]. Furthermore, this interaction was significantly enhanced in the presence of Ca^2+^/CaM and did not occur in the absence of Ca^2+^ [150]. In another in vitro ligation study, α-Syn was used as a bait to photo-crosslink possible interacting partners from bovine brain cytosol [80]: CaM was identified as the major cross-linked protein and the interaction was confirmed in SNKSH neuroblastoma cells. Binding of α-Syn to CaM could be blocked by CaM inhibitor calmidazol and was dependent on Ca^2+^ concentration. 

These studies strongly suggest that α-Syn interacts with principal players in the LTCC/CaMKII signal transduction pathway and that changes in its level of expression either during pathology or as a result of anti-synuclein therapy may have effects on the ability of neurons and other cells to adapt to the changing environmental and behavioral stimuli.

## 3. Conclusions and Future Directions

Although multiple studies implicate an involvement of α-Syn in transcription, it is unclear whether these pathways are physiologically relevant in humans. An important distinction needs to be drawn between studies of α-Syn overexpression/mutation, which are meant to emulate synucleinopathies, and those where the protein is depleted, which are intended to address its normal function. While several cellular regulators of transcription have been shown to mediate toxicity induced by α-Syn overexpression, it is unknown if decreasing α-Syn levels will have an opposite effect on gene expression. Similarly, even though reports of deficits in working and spatial memory in α-Syn null mice support a role of the protein in learning [56], further studies are required to determine the effects of therapeutic doses of anti-synuclein medications and more specific tests for the immediate early genes expression. Finally, while it is abundantly expressed in the brain, α-Syn is also expressed in the gut, liver, lungs and is necessary for the normal development and functioning of these tissues [152]. Silencing α-Syn in the periphery may cause side-effects by affecting the expression of genes in non-neuronal cells, a possibility that should be investigated in future studies.

## Figures and Tables

**Figure 1 genes-12-01166-f001:**
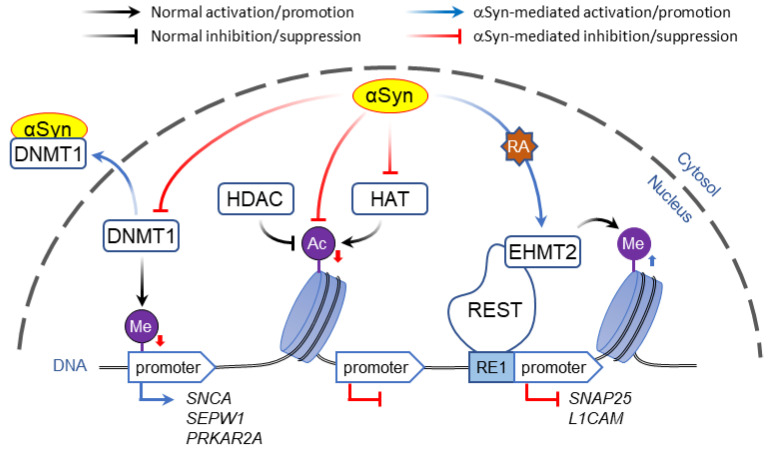
Role of α-Syn in post-translational modification of DNA and histones. α-Syn has been shown to alter PTM of DNA and histones via multiple mechanisms, three of which are shown here. Left, α-Syn can promote the translocation of DNMT1 from the nucleus to the cytosol leading to DNA hypomethylation and increased genes expression. Middle, α-Syn may interact directly with histone 3 or inhibit HAT activity; in both scenarios reduced histone acetylation leads to decreased gene expression, which can be rescued by HDAC inhibitors. Right, α-Syn may change histone methylation by activating EHMT2 in a retinoic acid (RA)-dependent manner, thus decreasing the expression of genes regulated by REST complex. Arrows next to PTM sites show the effect at higher levels of α-Syn.

**Figure 2 genes-12-01166-f002:**
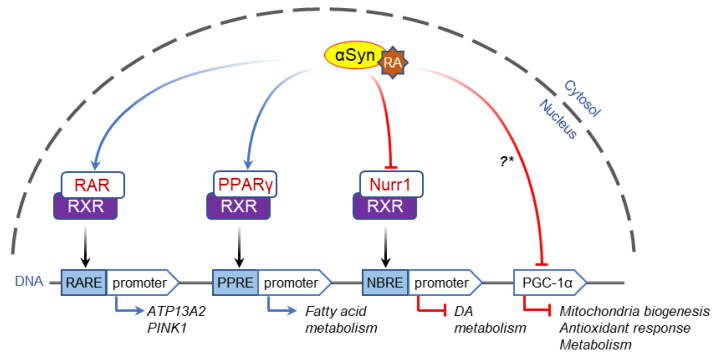
Effects of α-Syn on transcription factors. Left, α-Syn is translocated to the nucleus in a complex with retinoic acid (RA) where it interacts with and activates RAR and PPARγ nuclear receptors, followed by activation of their corresponding response elements on gene promoter regions. Middle, overexpression of α-Syn also induces downregulation of the orphan nuclear receptor Nurr1, decreasing expression of Death Receptor 5 dependent genes, including those involved in dopamine biosynthesis (*TH*, *AADC, GCH1*). Right, α-Syn binds the promoter region of PPARγ co-activator (PGC-1α) gene when overexpressed. This leads to the downregulation of PGC-1α transcription thus reducing cellular protection against oxidative stress and bioenergetic burden. *—It is currently unknown if the interaction with PGC-1α is RA-dependent.

**Figure 3 genes-12-01166-f003:**
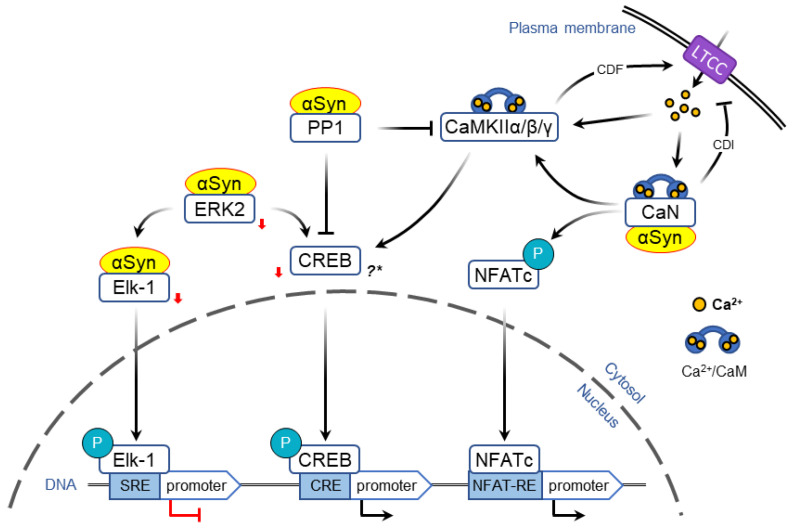
Role of α-Syn in regulation of immediate early gene responses. Left, Cytosolic α-Syn can interact with transcription factors involved in mitogen-activated protein kinase (MAPK) signaling such as ERK2 and Elk1, resulting in the downregulation of gene response elements. Middle, Activity-dependent pCREB activation is triggered by Ca^2+^ influx through LTCC which activates calmodulin (CaM), which then binds to and activates calcineurin (CaN). Activated CaM phosphorylates CaMKIIα/β, which in turn phosphorylates CaMKIIγ that results in the dissociation of Ca^2+^/CaM from CaMKIIγ and its translocation to the nucleus—the step that requires CaN activity. Nuclear Ca^2+^/CaM phosphorylates CREB via CaMKK and CaMKIV pathways, resulting in the binding of pCREB to its promoter on the DNA. Serine/threonine protein phosphatase 1 (PP1) inhibits this cascade by dephosphorylating pCREB and CaMKIIα/β. Right, Activated CaN dephosphorylates NFATc, exposing its nuclear localization sequences (NLS) and facilitating its translocation to the nucleus, where it binds to promoter DNA to control transcription. Binding of α-Syn to Ca^2+^/CaM/CaN complex and PP1 may affect the induction of NFATc and pCREB factors as well as the gating properties of LTCC by interfering with Ca^2+^-dependent facilitation (CDF) or inactivation (CDI). *—The effect of α-Syn expression on pCREB induction has not been investigated to date.

## Data Availability

Not applicable.

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
