# Peer review of "Roles for α-Synuclein in Gene Expression"

_genes, 2021, doi:10.3390/genes12081166_

Round 1

Reviewer 1 Report

The report by Somayaji et al describes a variety of potential functions of alpha synuclein that include many that impact cellular homeostasis. The principal argument against the many potential functions/activities of alpha synuclein is the benign phenotype of mice in which the gene has been deleted in mice, under laboratory conditions. Embryonic and post natal development, normal behavior, basal neuronal activity, food intake, growth,etc are not affected by the loss of alpha sunclein expression. A similar phenotype is seen in mice from which all three synuclein genes are deleted. The one exception is a report that argues for a difference in the response to amphetamine in the aS ko mice, however an alternative explanation implicates an effect of that construction on local  amphetamine pharmacokinetics. 

It has often been said that we can always do studies that describe "what can happen"  that are frequently distinctly different from "what does happen."  

Author Response

Roles for alpha-Synuclein in gene expression

We thank the reviewers for their valuable feedback on our manuscript and we hope that we addressed all of their concerns.

REVIEWER 1:

The report by Somayaji et al describes a variety of potential functions of alpha synuclein that include many that impact cellular homeostasis. The principal argument against the many potential functions/activities of alpha synuclein is the benign phenotype of mice in which the gene has been deleted in mice, under laboratory conditions. Embryonic and post-natal development, normal behavior, basal neuronal activity, food intake, growth, etc are not affected by the loss of alpha synuclein expression. A similar phenotype is seen in mice from which all three synuclein genes are deleted. The one exception is a report that argues for a difference in the response to amphetamine in the aS ko mice, however an alternative explanation implicates an effect of that construction on local amphetamine pharmacokinetics. 

It has often been said that we can always do studies that describe "what can happen" that are frequently distinctly different from "what does happen."  

We appreciate the reviewer’s concern about the physiological relevance of the studies in our manuscript. While we agree that the effects of constitutive aSyn deficiency are relatively subtle, we disagree with the assessment that mice lacking aSyn do not show a significant phenotype. In addition to  the  study that we believe the Reviewer is referring to from our group and collaborators that showed a reduction in striatal dopamine content and locomotor response in response to amphetamine in aSyn KO mice (PMID: 10707987), another study cited in the review demonstrates cognitive impairments in these animals (PMID:22469626), and we have also recently shown that aSyn KO mice have deficient short-term and long-term synaptic plasticity in the dopamine system in vivo (PMID: 33273122). Second, post-developmental silencing of aSyn has been shown to cause neuroinflammation and degeneration of nigral dopaminergic neurons (PMID:20551914, 21565333); it may be that aSyn KD results are particularly relevant to therapies aimed at aSyn reduced expression or presence than aSyn KO, which may be compensated for during development. This is not uncommon in constitutive knockout animals: for example, mice that completely lack SN DA neurons from birth (Pitx3 KO line) are blind but have a hyperactive motor phenotype. The lack of a parkinsonian phenotype in this line certainly does not mean that SN DA neurons are unimportant in PD. The main objective of our review is to highlight the physiological role of aSyn so that potential side effects caused by its deficiency, including during future therapeutic approaches, are considered.

Reviewer 2 Report

The review by Somayaji et al. explain the role of alpha-synuclein in synucleinopathies, fully exploring its involvement in influencing gene expression. The manuscript is properly systematic for direct interactions between and gene activity, describing the regulatory role in different ways, like in activation or down regulation of transcription factors, in post-translation modification of DNA and histones and in regulation of immediate early genes responses.

The review is a timely update on this important issue.

Minor comments

In the introductory part, however, in the brief excursus on synucleinopathies, it would be more interesting to discuss the cases that describe the interaction between misfolded alpha-synuclein and LRRK2 and parkin genes, rather than the cases where it does not occur.

More generally, the review could be enriched arguing the pathological role of alpha-synuclein and its influence on gene expression, since the article focused only on physiological state of the protein.

In “Therapeutic silencing of a-syn” chapter, there is a primary concept that needs to focalize. Antibody therapy is targeted against oligomerized and misfolded pathological alpha-synuclein but it is not involved on physiological monomeric forms. There are selective antibodies for oligomeric and protofibrils alpha-synuclein, as described in the papers by Tozzi et al., Biological Psychiatry, 2016 and Durante et al., Brain, 2019.

Author Response

Roles for alpha-Synuclein in gene expression

We thank the reviewers for their valuable feedback on our manuscript and we hope that we addressed all of their concerns.

REVIEWER 2:

The review by Somayaji et al. explain the role of alpha-synuclein in synucleinopathies, fully exploring its involvement in influencing gene expression. The manuscript is properly systematic for direct interactions between and gene activity, describing the regulatory role in different ways, like in activation or down regulation of transcription factors, in post-translation modification of DNA and histones and in regulation of immediate early genes responses. The review is a timely update on this important issue. 

We thank the reviewer for their kind comments.

Minor comments

In the introductory part, however, in the brief excursus on synucleinopathies, it would be more interesting to discuss the cases that describe the interaction between misfolded alpha-synuclein and LRRK2 and parkin genes, rather than the cases where it does not occur.

We had placed this discussion later in the text (lines 117), but agree that the paragraph about synucleinopathies is a better place and have transferred the references to Lines 71-74.

More generally, the review could be enriched arguing the pathological role of alpha-synuclein and its influence on gene expression, since the article focused only on physiological state of the protein. 

While the main focus of this review is to highlight the physiological role of aSyn and its endogenous effects in regulating gene expression, we agree with the critique and discuss the effect of pathological aSyn expression on gene expression in sections 2.1.2, 2.2. We also discuss problems associated with interpreting data from aSyn overexpression and aSyn deficiency studies in Conclusions and Future Directions section (Lines 396-405).

In “Therapeutic silencing of a-syn” chapter, there is a primary concept that needs to focalize. Antibody therapy is targeted against oligomerized and misfolded pathological alpha-synuclein but it is not involved on physiological monomeric forms. There are selective antibodies for oligomeric and protofibrils alpha-synuclein, as described in the papers by Tozzi et al., Biological Psychiatry, 2016 and Durante et al., Brain, 2019.

This is an excellent point, but note that some antibodies target both oligomeric and monomeric forms of aSyn. To better address the difference between approaches that deplete aSyn globally and those that only target pathogenic forms of the protein we have changed the relevant paragraph on page 3 that now reads (lines 91-94):

“With the possible exception of immunotherapies that target pathological oligomeric and misfolded forms of α-Syn, such as cinpanemab [51], prasinezumab [52] and ABBV-0805 (currently in phase I clinical trial), these approaches are expected to deplete monomeric α-Syn. As this protein is abundantly expressed in the brain and the periphery, its normal function and potential side effects of decreased α-Syn levels need careful evaluation.”

Reviewer 3 Report

This is an excellent review focused on the impact of alpha-synuclein on gene expression. The review is well-organized and good reading.  The topic is intriguing and offers a new angle for looking at both alpha-synuclein roles and alpha-synuclein based therapeutic strategies. Few minor points need to be addressed.

  1. Page 1, lines 35-36: the reference 9 is not right for assessing which are the known point mutations related to familial Parkinson’s disease. Please replace.
  2. Page 2, lines 49-52: please complete the sentences based on the evidence that Lewy bodies and Lewy neurites do not represent all the inclusions in synucleinopathies. In multiple-system atrophy the inclusions are Papp-Lantos bodies. In addition, Lewy bodies and Lewy neurites are intraneuronal aggregates whereas Papp-Lantos bodies are detected inside oligodendroglial cells. This could be mentioned.
  3. SNCA gene could be written in italic throughout the text.
  4. Two abbreviations for alpha-synuclein have been missed at page 9, line 379 and 391. 

Author Response

Roles for alpha-Synuclein in gene expression

We thank the reviewers for their valuable feedback on our manuscript and we hope that we addressed all of their concerns.

REVIEWER 3:

This is an excellent review focused on the impact of alpha-synuclein on gene expression. The review is well-organized and good reading.  The topic is intriguing and offers a new angle for looking at both alpha-synuclein roles and alpha-synuclein based therapeutic strategies. Few minor points need to be addressed.

We thank the reviewer for their kind comments.

  1. Page 1, lines 35-36: the reference 9 is not right for assessing which are the known point mutations related to familial Parkinson’s disease. Please replace.

Thank you. We have replaced the reference and now site two recent reviews on this topic: PMID: 27080380, 24262183 (line 36).

  1. Page 2, lines 49-52: please complete the sentences based on the evidence that Lewy bodies and Lewy neurites do not represent all the inclusions in synucleinopathies. In multiple-system atrophy the inclusions are Papp-Lantos bodies. In addition, Lewy bodies and Lewy neurites are intraneuronal aggregates whereas Papp-Lantos bodies are detected inside oligodendroglial cells. This could be mentioned.

Thank you. We have rephrased the sentence and included the details mentioned above (lines 52-57), which now reads:

“All “synucleinopathies” are characterized by the accumulation of insoluble aggre-gates mainly comprised of amyloid fibrils of phosphorylated α-Syn. In Parkinson’s Dis-ease (PD), PD with dementia, dementia with Lewy bodies and pure autonomic failure the inclusions (termed Lewy bodies and Lewy neurites) are found in peripheral and central neurons [20, 21], whereas in multiple system atrophy, the argyrophilic glial cytoplasmic inclusions (also known as Papp-Lantos bodies) are detected inside oligodendrocytes [22].”

  1. SNCA gene could be written in italic throughout the text.

We have fixed this throughout the text.

  1. Two abbreviations for alpha-synuclein have been missed at page 9, line 379 and 391

We have run a search for the term “synuclein” and are unable to find the missed abbreviation. Possibly the reviewer is referring to the term “anti-synuclein”, but the use of this term is intended as an antibody might react with additional all types of synucleins.

Reviewer 4 Report

The review "Roles of alpha-synuclein in gene expression" deals with putative important physiological functions of the disease-associated protein alpha-synuclein. This is of high interest, as down-regulation of this protein has been proposed for treating alpha-synucleinopathies, including Parkinson's disease, althouth the side effects of these approaches remain unclear. The authors focused on the role of alpha-synuclein in regulation the expression of genes via direct binding to chromatin (DNA and histones), regulation of nuclear receptors that affect gene expression, and modulating signaling pathways affecting gene expression.

The review is very well-written, easy to read with very illlustrative and meaningful figures.

Minor comments:

Please carefully go through the manuscript; there are some typos (e.g. line 42, aSyn; line 261, synucleonopathies) that should be eliminated before publication.

Author Response

Roles for alpha-Synuclein in gene expression

We thank the reviewers for their valuable feedback on our manuscript and we hope that we addressed all of their concerns.

REVIEWER 4:

The review "Roles of alpha-synuclein in gene expression" deals with putative important physiological functions of the disease-associated protein alpha-synuclein. This is of high interest, as down-regulation of this protein has been proposed for treating alpha-synucleinopathies, including Parkinson's disease, althouth the side effects of these approaches remain unclear. The authors focused on the role of alpha-synuclein in regulation the expression of genes via direct binding to chromatin (DNA and histones), regulation of nuclear receptors that affect gene expression, and modulating signaling pathways affecting gene expression. The review is very well-written, easy to read with very illlustrative and meaningful figures.

We thank the reviewer for their kind comments.

Minor comments:

Please carefully go through the manuscript; there are some typos (e.g. line 42, aSyn; line 261, synucleonopathies) that should be eliminated before publication.

Thank you. We have additionally proofread the manuscript for typos and grammar.
